# Long-term farming and cropping systems with contrasting nitrogen forms and input diversity influence soil prokaryotic diversity in the central highlands of Kenya

Susan Wairimu Muriuki[1], Anne Kambura[2], Julius Mugweru[1], Kennedy Wanjau[3], David Bautze[4], Thomas Dubois[5], Felix Matheri[5], Edwin Mwangi[5], Nancy Mwende[5], Edwin Ngetha[1], Edward Nderitu Karanja[5]*

**1** Department of Biological Sciences, School of Pure and Applied Sciences, University of Embu, Embu, Kenya, **2** School of Agriculture, Earth and Environmental Sciences, Taita Taveta University, Voi, Kenya, **3** International Livestock Research Institute, Nairobi, Kenya, **4** Research Institute of Organic Agriculture (FiBL), Frick, Switzerland, **5** International Centre for Insect Physiology and Ecology (icipe), Nairobi, Kenya

* ekaranja@icipe.org

## Abstract

### Background

Understanding how farming systems management influences soil microbial communities is essential for advancing sustainable agriculture in tropical regions. Long-term experiments provide valuable opportunities to assess how cumulative management practices shape soil microbial diversity and community composition.

### Methods

We investigated prokaryotic communities after 15 years of continuous management in the Long-Term Farming Systems Comparison Trial (SysCom-Kenya) at two contrasting sites (Chuka and Thika) in the Central Highlands of Kenya. Four systems were evaluated: conventional low-input (Conv-Low), conventional high-input (Conv-High), organic low-input (Org-Low), and organic high-input (Org-High). Soil samples were collected at key crop growth stages (vegetative, reproductive, and maturity) of maize, baby corn, and potato. Prokaryotic diversity and community composition were characterized using 16S rRNA gene amplicon sequencing, and soil chemical properties were analyzed to explore potential abiotic drivers.

### Results

Prokaryotic community composition and diversity varied primarily with site and farming system, with secondary variation across crop growth stages. Across all systems, communities were dominated by members of the phyla Proteobacteria and Actinobacteria, followed by Acidobacteria, Firmicutes, and Chloroflexi. Organic systems, particularly organic high-input, tended to support higher richness and evenness than

**Data availability statement:** The demultiplexed high-quality sequence reads are deposited in the National Center for Biotechnology Information (NCBI) Sequence Read Archive (SRA) under BioProject ID: PRJNA1227412, available at: https://www.ncbi.nlm.nih.gov/sra/PRJNA1227412. Metadata, soil chemistry data, QIIME input files, and R analysis scripts are deposited in Zenodo at https://doi.org/10.5281/zenodo.14940274 and https://doi.org/10.5281/zenodo.14940273.

**Funding:** The authors gratefully acknowledge the financial support for this research by the following organizations and agencies: Biovision Foundation (Grant No: 1040), Coop Sustainability Fund (Grant No: 1040), Liechtenstein Development Service (LED) (Grant No: 1040), and the Swiss Agency for Development and Cooperation (SDC) (Grant No: 1040); the Swedish International Development Cooperation Agency (Sida); the Swiss Agency for Development and Cooperation (SDC); the Australian Centre for International Agricultural Research (ACIAR); the Government of Norway; the German Federal Ministry for Economic Cooperation and Development (BMZ); and the Government of the Republic of Kenya. The views expressed herein do not necessarily reflect the official opinion of the donors.

**Competing interests:** The authors declare that they have no competing interests.

conventional systems, while low-input systems consistently exhibited lower prokaryotic richness and diversity than high-input systems. Diversity generally increased toward later crop growth stages, although phenological effects were variable. Canonical correspondence analysis identified soil pH, ammonium-N, and available phosphorus as important correlates of community structure, especially at the drier Thika site. Taxon-specific enrichment patterns differed across systems and crop stages, indicating compositional differentiation rather than functional dominance.

## Conclusion

Our findings indicate that long-term management intensity and organic input diversity exert a stronger influence on soil prokaryotic communities than short-term crop phenology. Despite limitations from sample pooling, this study provides novel evidence from sub-Saharan Africa that diversified organic input management can enhance soil microbial diversity and potential resilience, supporting sustainable soil management in tropical farming systems.

## Introduction

Soil microorganisms are fundamental components of terrestrial ecosystems, driving processes that underpin soil fertility, crop productivity, and ecosystem resilience. Through their roles in organic matter decomposition, nutrient mineralization, soil aggregation, and the regulation of greenhouse gas fluxes, soil prokaryotes strongly influence the sustainability of agricultural systems, particularly in tropical smallholder contexts where soils are often degraded and nutrient-limited [1–4]. Farming practices, including crop rotations, fertilizer regimes, and organic input management, are therefore key determinants of soil microbial community structure and function [5–7].

Nitrogen (N) management is a major driver of soil microbial dynamics because microbial communities mediate N transformations such as ammonification, nitrification, and denitrification [8,9]. However, in practice, N management in sub-Saharan Africa (SSA) farming systems rarely involves a single N form. Instead, cropping systems often receive combinations of organic and inorganic inputs, differing not only in N form but also in input diversity, timing, and total nutrient load. These factors interact to shape microbial habitats and resource heterogeneity, with important implications for microbial diversity and stability [10,11]. In SSA, declining soil fertility remains a major constraint to agricultural productivity, with many rainfed systems producing yields far below their potential [12,13]. Organic and agroecological farming approaches have been proposed as pathways to restore soil health and microbial diversity, yet evidence from long-term field experiments in SSA remains limited. Moreover, results from existing studies are sometimes contrasting, highlighting the need for long-term perspectives that capture cumulative management effects. The Long-Term Farming Systems Comparison Trial (SysCom-Kenya) provides a unique platform to examine these dynamics. An earlier study from this trial by [14] reported higher prokaryotic diversity in conventional systems compared to organic systems

during earlier phases of the experiment. In contrast, emerging evidence from long-term trials suggests that prolonged application of diverse organic inputs can enhance microbial richness, evenness, and resilience over time. Differences in study timing, crop rotations, sampling seasons, and analytical pipelines may also contribute to contrasting outcomes, underscoring the importance of reassessing microbial responses after extended periods of continuous management.

Beyond farming system type, organic input diversity including composts, green manures, mulches, plant teas, and legume intercrops may play a critical role in shaping soil microbial communities by increasing substrate heterogeneity and niche availability. In the SysCom-Kenya trial, while all systems receive farmyard manure, organic systems receive a wider diversity of organic amendments, and high-input systems receive substantially higher total nutrient inputs over the crop rotation. These differences may be as influential as nitrogen form alone in determining microbial community structure.

Crop phenology further modulates soil microbial dynamics. Root exudation, rhizodeposition, and residue inputs change across growth stages, potentially leading to temporal shifts in microbial diversity and composition. However, the relative importance of crop developmental stage versus long-term system management remains poorly understood under tropical field conditions, particularly in African smallholder systems.

Based on ecological theory and emerging evidence from long-term farming systems, we hypothesized that: (i) farming systems receiving more diverse and higher-intensity organic inputs would support higher prokaryotic richness and evenness than low-input and conventional systems; (ii) long-term system management effects would outweigh short-term crop phenological effects in structuring prokaryotic communities; and (iii) soil chemical properties associated with nutrient availability and habitat suitability, particularly pH, ammonium-N, and available phosphorus, would be key correlates of prokaryotic community composition, with site-specific differences reflecting contrasting climatic conditions.

In this study, we investigated soil prokaryotic communities after 15 years of continuous management in the SysCom-Kenya trial at two contrasting sites (Chuka and Thika). Using 16S rRNA gene amplicon sequencing and soil chemical analyses, we aimed to:

i.  compare prokaryotic diversity and community composition across farming systems differing in input intensity and organic input diversity;

ii.  assess temporal shifts in prokaryotic communities across crop developmental stages; and

iii.  identify key soil chemical drivers associated with observed community patterns.

By revisiting this long-term experiment, our study provides rare insight into how prolonged management intensity and organic input diversity interact with crop phenology and site conditions to shape soil microbial communities in tropical agroecosystems.

## Materials and methods

### Study site

The long-term trials were established in 2007 under the Farming Systems Comparison in the Tropics (SysCom) program of the Research Institute of Organic Agriculture (FiBL). The trials are located in the sub-humid Central Highlands of Kenya at Chuka (0° 20.864' S, 37° 38.792' E) and Thika (1° 0.231' S, 37° 04.747' E). Chuka lies in the Upper Midland 2 (UM2) agro-ecological zone at an elevation of 1,458 m above sea level and receives a mean annual rainfall of approximately 1,050 mm, while Thika is in the Upper Midland 3 (UM3) zone at approximately 1,500 m above sea level with an average annual rainfall of about 900 mm. Both sites experience a bimodal rainfall pattern with long and short rainy seasons. Mean annual temperatures at both sites are approximately 20 °C (range: 15–27 °C). Soils at Chuka are classified as Humic Nitisols, while those at Thika are Rhodic Nitisols, both characterized by high clay contents typical of deeply weathered tropical soils [15,16].

 

## Farming systems

Four farming systems were evaluated: Conventional Low Input (Conv-Low), Conventional High Input (Conv-High), Organic Low Input (Org-Low), and Organic High Input (Org-High). High-input systems represent market-oriented production with supplemental irrigation during dry spells, while low-input systems reflect rainfed smallholder farming conditions. Crop choices and rotations followed locally relevant practices and were consistent with previous SysCom Kenya studies [17,18].

Although all systems received farmyard manure (FYM), organic systems differed from conventional systems by receiving a greater diversity of organic inputs, including compost, *Tithonia diversifolia* mulch, plant teas, and legume intercrops. In addition, high-input systems received substantially higher total nutrient inputs over the crop rotation and supplemental irrigation compared to low-input systems. Thus, systems differed not only in nitrogen form but also in input diversity, timing, and total nutrient load.

## Experimental design

At each site, the four farming systems were arranged in a randomized complete block design (RCBD) with four replicates. Each experimental plot measured 8 × 8 m, with a net plot size of 6 × 6 m used for sampling and data collection. The experiment follows a long-term, two-season, three-year crop rotation framework as previously described by [19].

## Crop rotation and field management

During the year 2021 cropping, both the long and short rainy seasons were utilized. In the high-input systems, babycorn (*Zea mays* var. Pan 14) intercropped with greenleaf desmodium (*Desmodium intortum*) was grown during the long rains, followed by potato (*Solanum tuberosum* var. Shangi) intercropped with dolichos (*Lablab purpureus*) and *Desmodium intortum* during the short rains. In the low-input systems, maize (*Zea mays* var. H513) intercropped with common bean (*Phaseolus vulgaris* var. GLP 92) was grown during the long rains, followed by potato intercropped with *Lablab purpureus* in the short rains.

Tillage practices followed local smallholder farming methods and consisted of shallow conventional tillage using hand hoes. No deep inversion tillage was applied. These practices are representative of the dominant management systems in the Central Highlands of Kenya.

## Nutrient management

Nutrient management strategies varied by system but were designed to supply equivalent levels of nitrogen (N) and phosphorus (P) across conventional and organic systems within the same input category. In the Conv-Low system, decomposed farmyard manure (FYM) was supplemented with 50 kg ha⁻¹ of diammonium phosphate (DAP) during the long rains and 100 kg ha⁻¹ during the short rains, providing total N and P inputs of 45 and 27 kg ha⁻¹ (long rains), and 45 and 50 kg ha⁻¹ (short rains), respectively. The Org-Low system used the same crop combinations as Conv-Low but received nutrient inputs from decomposed FYM, *Tithonia diversifolia* mulch, and rock phosphate (RP) applied at 100 kg ha⁻¹ (long rains) and 200 kg ha⁻¹ (short rains). These applications were calibrated to match the total N and P levels applied in the Conv-Low system. In the Conv-High system, nutrient inputs included decomposed FYM supplemented with 200 kg ha⁻¹ DAP and 100 kg ha⁻¹ calcium ammonium nitrate (CAN) in the long rains, and 300 kg ha⁻¹ triple super phosphate (TSP) along with 200 kg ha⁻¹ CAN in the short rains. These inputs resulted in total N and P applications of 113 and 60 kg ha⁻¹ (long rains), and 90 and 100 kg ha⁻¹ (short rains), respectively. The Org-High system employed compost derived from the same quantity of fresh FYM used in Conv-High, supplemented with *Tithonia diversifolia* mulch and/or plant tea, and RP applied at rates of 364 kg ha⁻¹ (long rains) and 581 kg ha⁻¹ (short rains). These rates were selected to match the total N and P input levels of the Conv-High system. In all organic systems, *Tithonia*

diversifolia mulch was applied shortly after germination to serve as an early source of readily available nitrogen. Phosphorus was applied at planting and during the vegetative growth stage, while nitrogen was applied in split doses at planting, vegetative, and reproductive stages. All synthetic fertilizer application rates were based on regional recommendations provided by [20]. Pest and disease management practices were informed by bi-weekly scouting. Conventional systems employed synthetic pesticides and fungicides for control, while organic systems relied on certified biological products available in the local market.

## Soil sampling

Soil samples were collected at key physiological growth stages of each crop. For babycorn and maize, samples were taken at vegetative, tasseling/silking, grain formation, and maturity stages. For potato, sampling occurred at vegetative, flowering, and maturity stages. Within each plot, 12 soil cores were collected in a zigzag pattern from the topsoil (0–20 cm) using an Edelman auger and composited. For soil chemical analyses, replicate-level sampling was maintained. For microbial analyses, replicate samples within each farming system, growth stage, and site were pooled to generate a single composite sample. This pooling strategy was adopted to reduce sequencing costs and to identify broad system-level patterns across the long-term experiment.

Soil samples for molecular analysis were immediately preserved on dry ice in cooler boxes and transported to the International Centre of Insect Physiology and Ecology (icipe), Nairobi, for storage at − 80 °C until nucleic acid extraction. Samples designated for chemical analysis were also preserved on dry ice and delivered on the same day to Crop Nutritional Laboratory Services (CNLS), Nairobi, for analysis.

## Soil chemical analyses

At CNLS, soil samples were analyzed for key chemical properties using standard protocols. Soil pH was measured potentiometrically as described by [21]. Nitrate nitrogen ($NO_3^-$-N) and ammonium nitrogen ($NH_4^+$-N) were determined spectrophotometrically following the procedures of [22] and [23], respectively. Total nitrogen was quantified using the Kjeldahl method [24], while total phosphorus was determined calorimetrically [21]. Available phosphorus was extracted using the Olsen method, also according to [21]. All soil chemical variables were analyzed at the replicate level. Soil organic carbon was not measured in this study since the focus was mainly on N-mineralization.

## Prokaryotic DNA extraction and 16S rRNA gene sequencing

Total genomic DNA was extracted using the PureLink™ Microbiome DNA Purification Kit (Thermo Fisher Scientific, USA) following the manufacturer's protocol. DNA quality, purity, and concentration were assessed using a NanoDrop™ 2000 spectrophotometer (Thermo Fisher Scientific, USA) and agarose gel electrophoresis. To improve DNA extraction from the nitisols soil which are known to have high clay content, mechanical disruption was applied which involved using bead beating tubes to increase the recovery of DNA in the soil. The extracted DNA samples were shipped to MR DNA Laboratory (Shallowater, Texas, USA) for high-throughput sequencing of the V4 - V5 hypervariable regions of the 16S rRNA gene. Amplification was performed using universal primers 515F (5′-GTGCCAGCMGCCGCGGTAA-3′) and 806R (5′-GGACTACHVGGGTWTCTAAT-3′), which amplify both bacterial and archaeal taxa [25]. PCR amplification was conducted using the HotStarTaq® Plus Master Mix Kit (Qiagen, USA), and amplicons were purified using calibrated AMPure® XP beads (Beckman Coulter Life Sciences, USA). Sequencing libraries were prepared following Illumina's protocols and sequenced on the Illumina MiSeq platform (2 × 250 bp paired-end chemistry). Quality control was performed at MR DNA Laboratory. Raw reads were demultiplexed, and low-quality sequences were removed using an average Phred score cutoff of 30. Adapter and primer sequences were trimmed and reads with ambiguous bases or shorter than 150 bp after trimming were discarded. Paired-end reads were merged with a minimum overlap of 20 bp, and chimeric sequences were identified and removed using the UCHIME algorithm implemented in USEARCH. For downstream analyses, rarefaction

 

was applied to normalize sequencing depth across samples, ensuring that observed differences in prokaryotic diversity and community composition were not biased by uneven sequencing effort.

## Statistical analysis of soil chemical parameters

All soil chemical parameters were first tested for normality using the Shapiro – Wilk test. Parameters that met the assumptions of normality were subjected to analysis of variance (ANOVA) to assess the effects of farming system. Analyses were conducted separately for each crop development stage using the agricolae package in R version 4.3.1. Post hoc comparisons among the four farming systems (Conventional High, Organic High, Conventional Low, Organic Low) were performed using Tukey's Honest Significant Difference (HSD) test to determine differences in mean values of soil parameters. The analysis focused on soil pH, total nitrogen (N), nitrate nitrogen ($NO_3^-$-N), ammonium nitrogen ($NH_4^+$-N), Olsen phosphorus (available P), and total phosphorus (total P), revealing stage-specific trends across farming systems.

## Bioinformatic analysis of 16S rRNA sequence data

Raw sequence data were processed using ampliseq v2.7.0 [26]. Sequence quality was assessed using FastQC and summarized with MultiQC [27]. Primer sequences were trimmed using Cutadapt and reads lacking primer sequences were discarded. Forward and reverse reads were retained at lengths of 232 bp and 231 bp, respectively. Reads with homopolymer runs exceeding six base pairs were removed, and all reads were truncated at 324 bp. Demultiplexed high-quality sequences were submitted to the NCBI Sequence Read Archive (SRA) under BioProject ID PRJNA1227412, accessible at: https://www.ncbi.nlm.nih.gov/sra/PRJNA1227412. Amplicon sequence variant (ASV) inference was performed using the DADA2 v1.20.0 workflow [28], which provides finer-resolution and more reproducible community profiles than OTU clustering by correcting sequencing errors and avoiding arbitrary thresholds. Filtering was conducted using the filterAndTrim function [29], and error rates were learned for each sample. Reads were dereplicated using the derepFastq function, followed by ASV inference and abundance estimation. Taxonomic classification of ASVs was conducted using the assignTaxonomy function with the SILVA v138 reference database (downloaded from Zenodo). SILVA classifications were combined with RefSeq-RDP annotations using the cbind function. ASVs not classified at the genus level were further aligned using the AlignSeqs function [30]. The resulting taxonomy table was merged with the ASV abundance table. Taxa prevalence was calculated from absolute abundance data to identify dominant taxa across sampling stages. Taxonomic sub-setting at the phylum level was performed using the subset function, and bar plots of the top 20 most abundant prokaryotic classes were generated using ggplot2 v3.3.5. Statistical analyses began with Shapiro – Wilk tests for normality [31]. Kruskal – Wallis rank sum tests were then used to detect significant differences in soil variables across sampling stages and farming systems (adjusted $p < 0.01$), followed by Tukey's post hoc tests [32] for pairwise comparisons. Correlations among soil chemical variables were assessed using Pearson correlation coefficients via the Hmisc v4.5 package [32]. Significant correlations (adjusted $p < 0.01$) were visualized using the corrplot v0.9 package [33]. Alpha and beta diversity metrics were calculated using the vegan v2.5.7 [34] and phyloseq v1.16.2 [35] packages in R. Shannon diversity indices were used to evaluate alpha diversity across farming systems and sampling stages. Beta diversity was calculated using centered log-ratio (CLR)-transformed ASV tables with Bray – Curtis dissimilarity, implemented via the vegdist function [36]. Ordination was performed using principal coordinate analysis (PCoA) [37], and community dissimilarity was assessed using PERMANOVA with 999 permutations [37]. Environmental drivers of prokaryotic community structure were identified using redundancy analysis (RDA) [38]. Soil chemical data were standardized (z-scores) and checked for multicollinearity using the vif function from the car v3.0.11 package [39]. Significant explanatory variables were selected using forward stepwise selection (ordistep function, 1,000 permutations). Variables with variance inflation factor (VIF) > 10 were excluded. Model significance and contributions of individual predictors were evaluated using permutation-based ANOVA (1,000 permutations) [40]. Differentially abundant taxa between sampling stages were identified using the Linear Discriminant Analysis Effect Size (LEfSe) algorithm [41], implemented via the Microbiome Marker package in RStudio. Differences

were assessed using the Kruskal-Wallis test (adjusted $p < 0.01$), followed by pairwise Wilcoxon rank-sum tests [42]. Taxa with LDA scores ≥ 2 were considered significantly discriminative.

## Results

### Climate conditions during the study period

Rainfall and relative humidity (RH) exhibited marked spatial and seasonal variability across sites, whereas temperature remained relatively stable (S1 Fig). Chuka received higher cumulative rainfall than Thika during both the long rains (529 vs. 289 mm) and short rains (499 vs. 193 mm). Relative humidity followed rainfall patterns, with higher values at Chuka than Thika. Mean daily temperatures were similar at both sites, generally ranging between 16 and 21 °C. *Note:* Climatic data are presented to provide environmental context and are therefore reported in the Supporting Information.

### Sequencing depth, diversity and general characteristics of prokaryotic communities

Rarefaction curves for both sites approached saturation (S2 Fig), indicating sufficient sequencing depth to capture the dominant prokaryotic taxa across farming systems and crop growth stages. Although soil samples were pooled by farming system and crop growth stage to accommodate resource limitations, the observed ASV richness and diversity patterns therefore reflect system-level community characteristics rather than within- system variability.

High-quality 16S rRNA gene sequencing generated over 4.5 million reads across sites and farming systems, with comparable sequencing depth among systems (Table 1). At Chuka, the number of high-quality sequences ranged from 514,684–597,124 per system, while at Thika sequence counts ranged from 550,971–583,883, indicating minimal bias in sequencing effort across systems and sites.

Observed amplicon sequence variant (ASV) richness varied across farming systems and sites. At Chuka, observed ASVs ranged from 302 in the conventional low-input system to 462 in the organic high-input system. At Thika, richness ranged from 497 ASVs in the conventional low-input system to 543 ASVs in the organic high-input system. Across both sites, organic systems particularly the organic high-input system tended to exhibit higher ASV richness than conventional systems, while low-input systems consistently showed lower richness than their high-input counterparts.

**Table 1. High-quality 16S rRNA sequence distribution, observed ASVs and diversity indices across farming systems at Chuka and Thika sites.**

| Site | Farming system | High quality sequences | Observed ASVs | Shannon | Simpson | Fisher | Phylum | Class | Most abundant at phylum level |
|------|----------------|------------------------|---------------|---------|---------|--------|--------|-------|-------------------------------|
| Chuka | Conv-High | 514684 | 352 | 3.17 | 0.82 | 41.62 | 34 | 58 | Actinobacteria, Proteobacteria |
| Chuka | Conv-Low | 597124 | 302 | 3.28 | 0.84 | 34.86 | 34 | 58 | Actinobacteria, Acidobacteria |
| Chuka | Org-High | 596290 | 462 | 4.40 | 0.94 | 56.00 | 34 | 60 | Actinobacteria, Acidobacteria |
| Chuka | Org-Low | 540333 | 399 | 3.80 | 0.89 | 48.20 | 33 | 60 | Actinobacteria, Proteobacteria |
| Thika | Conv-High | 550971 | 528 | 4.13 | 0.93 | 64.62 | 35 | 58 | Actinobacteria, Proteobacteria |
| Thika | Conv-Low | 583883 | 497 | 3.98 | 0.91 | 60.82 | 33 | 58 | Actinobacteria, Gemmatimonadetes |
| Thika | Org-High | 574102 | 543 | 4.69 | 0.97 | 67.31 | 34 | 60 | Actinobacteria Acidobacteria |
| Thika | Org-Low | 561103 | 520 | 4.14 | 0.92 | 64.48 | 34 | 59 | Actinobacteria, Proteobacteria |

The table presents the number of high-quality sequences, amplicon sequence variants (ASVs), Shannon diversity index, Simpson index, and Fisher's alpha diversity. Data are shown for conventional high input (Conv-High), conventional low input (Conv-Low), organic high input (Org-High), and organic low input (Org-Low) systems.

Patterns in alpha diversity indices mirrored trends in ASV richness. Shannon diversity values were lowest in conventional systems at both sites (3.17–3.28 at Chuka; 3.98–4.13 at Thika) and highest in organic high-input systems (4.40 at Chuka; 4.69 at Thika). Simpson indices similarly indicated greater community evenness in organic systems, with the highest values observed in organic high-input systems (0.94 at Chuka; 0.97 at Thika). Fisher's alpha followed the same pattern, with higher values in organic and high-input systems across both sites.

Across all systems and sites, prokaryotic communities were taxonomically diverse, spanning 33–35 phyla and 58–60 classes. Community composition was consistently dominated by Actinobacteria and Proteobacteria, with Acidobacteria also prominent in several systems, particularly within organic systems.

## Relationships between soil chemical variables and community composition

Canonical correspondence analysis (CCA) revealed site-specific relationships between prokaryotic community composition and soil chemical variables. All measured soil variables (pH, $NH_4^+$-N, $NO_3^-$-N, total N, available P, and total P) were initially included in ordination analyses (S1 Table A-F). Following multicollinearity screening and forward selection, pH, ammonium-N, and available phosphorus were retained in the final models.

At Chuka, the CCA explained a small proportion of community variation, and associations between soil variables and community composition were weak (Fig 1A). At Thika, the retained variables explained a larger proportion of variation, with clearer alignment between pH, ammonium-N, available P, and community structure (Fig 1B).

## Abiotic drivers of prokaryotes diversity at different sampling stages and seasons

Soil pH, nitrate nitrogen, ammonium nitrogen, total nitrogen, total phosphorus, and available phosphorus were evaluated to determine which abiotic drivers significantly affected prokaryotic diversity at different sampling stages. At Chuka, CCA results suggested weak associations between prokaryotic community composition and the measured abiotic factors across both seasons (babycorn / maize season 1: $p = 0.831$, $R^2 = 0.0286$; potato season 2: $p = 0.184$, $R^2 = 0.0239$). In contrast, at Thika, pH, ammonium-N, and available phosphorus were more strongly aligned with community differences

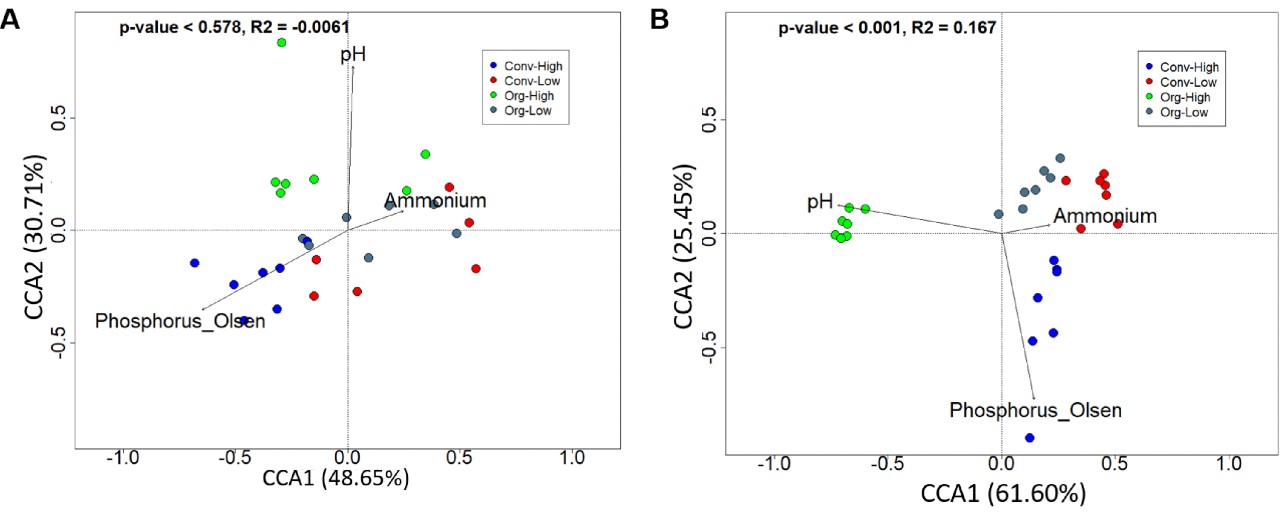

**Fig 1. Canonical correspondence analysis (CCA) of soil prokaryotic communities in relation to soil chemical properties.** Ordination plots show the relationships between soil variables and prokaryotic community composition at (A) Chuka and (B) Thika sites. The length and direction of arrows represent the strength and orientation of correlations with individual soil parameters. The points of different colors represent the farming systems.

at various growth stages in both seasons (babycorn / maize season 1: $p<0.001$, $R^2=0.2184$; potato season 2: $p<0.001$, $R^2=0.1868$) (Fig 2).

## Soil prokaryotic community composition in farming systems

The relative taxonomic abundance of dominant prokaryotic phyla differed across farming systems, seasons, and crop growth stages (Fig 3). Proteobacteria and Actinobacteria were the most dominant phyla across all treatments, accounting for approximately 50–70% of total relative abundance. Acidobacteria, Firmicutes, and Chloroflexi contributed to smaller

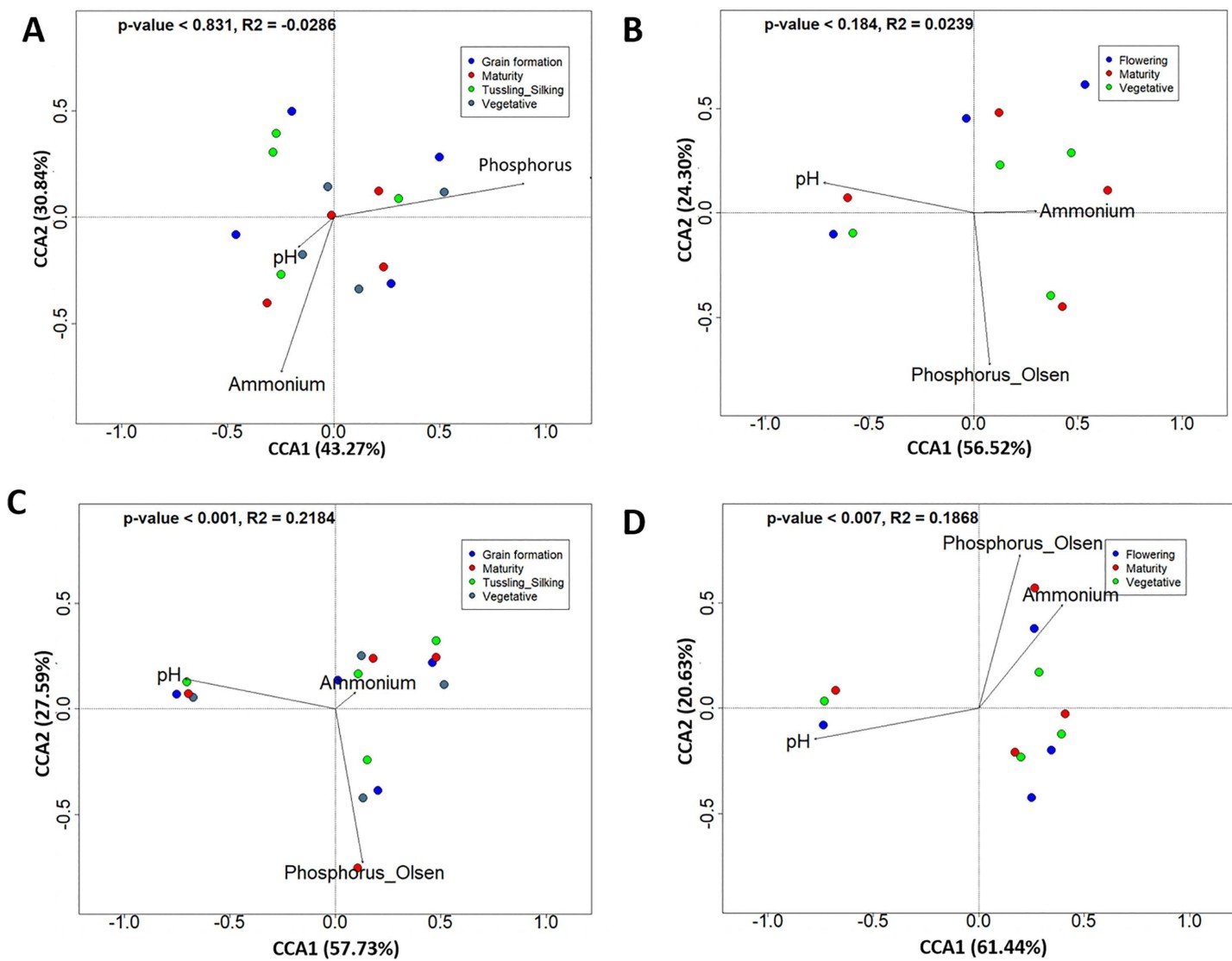

**Fig 2. Canonical correspondence analysis (CCA) of soil prokaryotic communities across crop growth stages and seasons.** Ordination plots show relationships between soil chemical properties and prokaryotic community composition at different crop growth stages. Panels (A) and (B) correspond to the Chuka site (season 1 and season 2), and panels (C) and (D) correspond to the Thika site (season 1 and season 2). Arrows represent the direction and strength of correlations with individual soil variables.

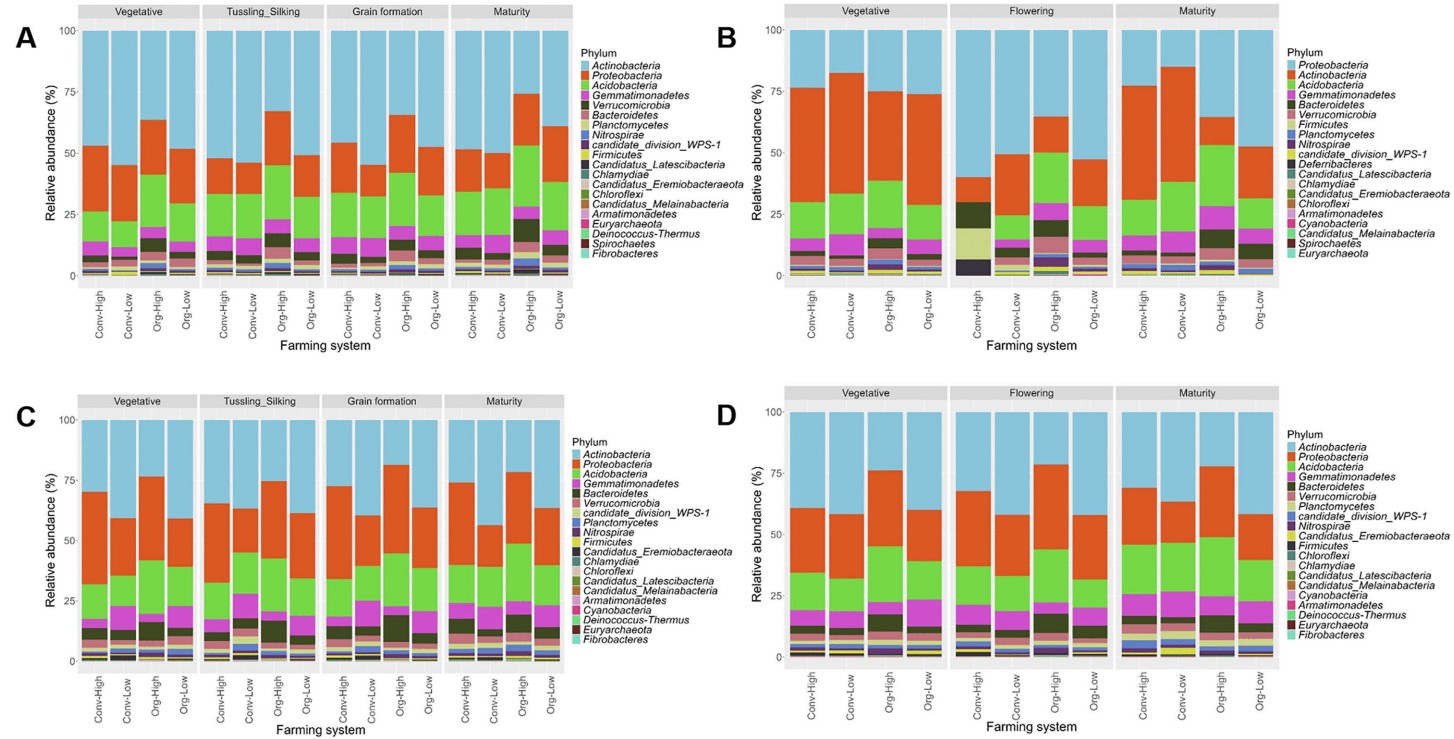

**Fig 3. Relative abundance of dominant prokaryotic phyla across farming systems and sites.** Stacked bar plots illustrate the relative abundance of the most predominant phyla in different farming systems. Panels (A) and (B) show the Chuka site for season 1 and season 2, respectively; panels (C) and (D) show the Thika site for season 1 and season 2, respectively.

proportions and showed greater variability across systems and stages. At the Chuka site, Proteobacteria and Actinobacteria dominated across all crop growth stages and farming systems in both seasons. During babycorn / maize season (Fig 3A), Actinobacteria was particularly abundant during the vegetative and tillering stages (tussling and silking), while Proteobacteria showed increased relative abundance during maturity. Potato season (Fig 3B) maintained a similar pattern, although Firmicutes and Acidobacteria exhibited greater variability among farming systems. At the Thika site, similar dominant phyla were observed. In babycorn / maize season (Fig 3C), Proteobacteria consistently dominated all systems and crop growth stages. In potato season (Fig 3D), the organic farming systems supported relatively higher levels of Firmicutes and Acidobacteria compared to conventional farming systems.

## Alpha diversity across farming systems

Patterns of α-diversity varied across farming systems and sites (Fig 4). At Chuka during the babycorn/maize season, prokaryotic communities varied significantly (Observed, Shannon and Simpson $p = 0.002$, 0.001 and 0.001 respectively) (Fig 4A). Additionally, organic systems generally showed higher observed ASVs, and diversity indices compared to conventional systems. Org-High and Org-Low reached 500 and 480 observed ASVs, respectively, while Conv-High averaged 300 and Conv-Low was below 300. Shannon diversity was also higher in organic systems (3.8–4.2) than in conventional (3.2–3.4), with Simpson's index following a similar trend (0.96 vs. 0.84–0.86). This pattern was broadly consistent in the potato season (Fig 4B), prokaryotic communities varied significantly for Shannon and Simpson $p = 0.03$ and 0.05 respectively in contrast to Observed which $p = 0.63$, where organic systems again maintained higher richness and evenness.

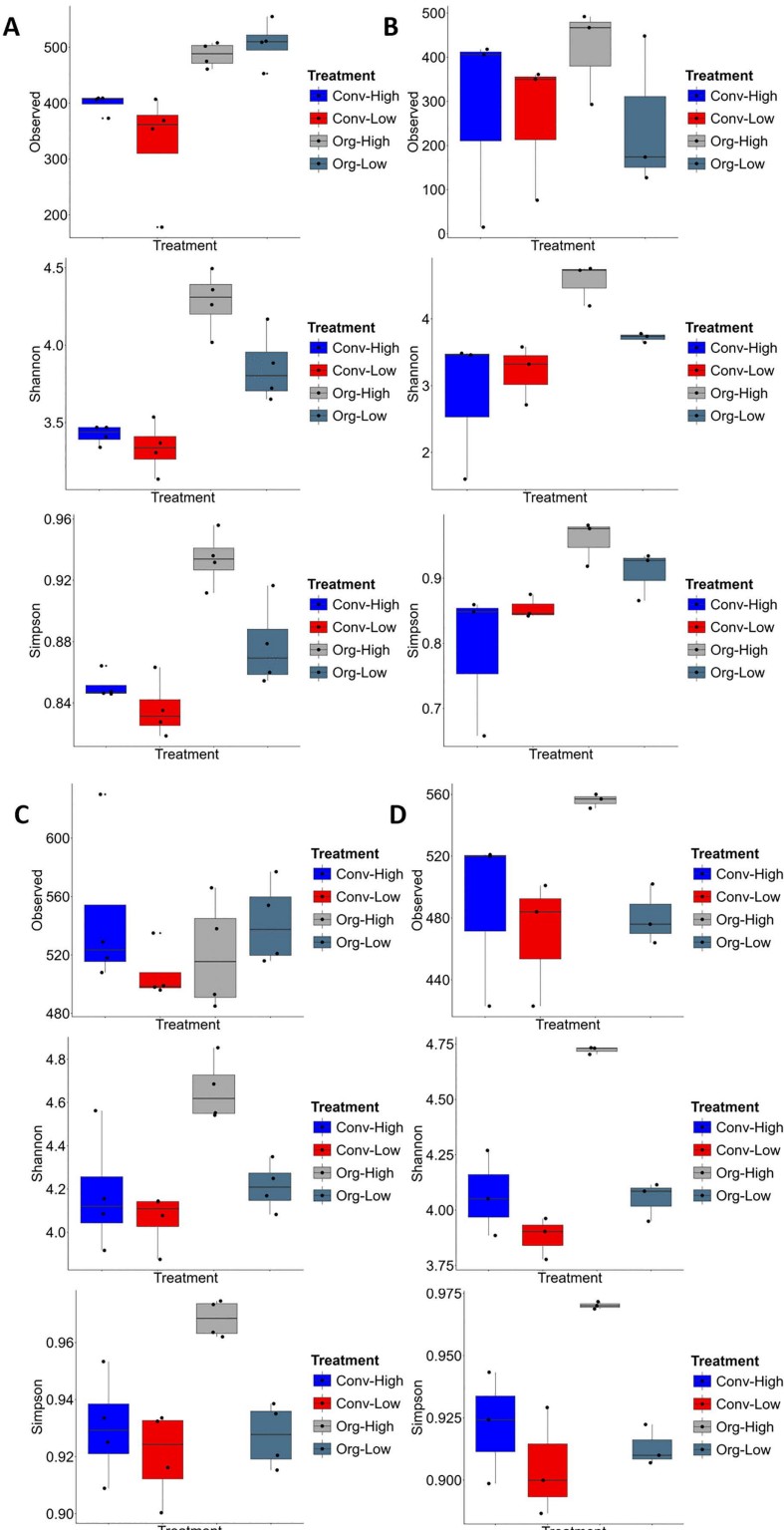

**Fig 4. Alpha diversity of soil prokaryotic communities across the different farming systems at Chuka and Thika sites.** Boxplots show observed amplicon sequence variants (ASVs), Shannon index, and Simpson index for farming systems at Chuka (A, season 1; B, season 2) and Thika (C, season 1; D, season 2). Higher values indicate greater richness and evenness of prokaryotic communities.

At Thika prokaryotic communities varied significantly for Observed, Shannon and Simpson $p = 0.002$, 0.002 and 0.001 respectively (Fig 4C). Additionally, prokaryotic communities varied significantly for Shannon and Simpson $p = 0.05$ and 0.004 respectively in contrast to Observed $p = 0.06$ (Fig 4D). Organic systems also supported higher richness and diversity. Org-High and Org-Low reached 560 and 540 observed ASVs, respectively, compared to 480 in Conv-Low. Shannon diversity was highest in Org-High (4.6) and lowest in Conv-Low (4.1). The Simpson index indicated greater evenness in organic systems (0.97) compared to conventional (0.91) and these differences persisted across both seasons.

## Alpha diversity across crop growth stages

Across sites and systems, alpha diversity metrics tended to increase from early vegetative stages toward grain formation and maturity (Fig 5). This trend was more consistently observed in organic systems than in conventional systems. However, separation among growth stages was not uniform across all farming systems, and variability among stages remained high.

At Chuka (Fig 5A), prokaryotic diversity did not vary significantly across cereal crop growth stages (Observed ASVs, Shannon, and Simpson indices; $p = 0.75$, 0.72, and 0.70, respectively). Nevertheless, richness and diversity tended to peak during the grain formation and maturity stages in baby corn/maize, with observed ASVs ranging from 430 to 450 and Shannon diversity values reaching up to 4.1. Earlier stages, including vegetative and tasseling/silking, were characterized by lower richness (350–380 ASVs) and lower diversity (Shannon index 3.5–3.7).

Similarly, during potato season at Chuka (Fig 5B), prokaryotic diversity did not differ significantly across crop growth stages (Observed ASVs, Shannon, and Simpson indices; $p = 0.12$, 0.56, and 0.35, respectively). Maturity showed the highest diversity across all metrics (Observed ASVs = 480; Shannon index = 4.2; Simpson = 0.93), while flowering stages consistently exhibited lower richness and diversity.

At Thika (Fig 5C and 5D), prokaryotic diversity also did not vary significantly across cereal crop growth stages (Observed ASVs, Shannon, and Simpson indices; $p = 0.22$, 0.69, and 0.80, respectively) or across potato growth stages (Observed ASVs, Shannon, and Simpson indices; $p = 0.54$, 0.50, and 0.55, respectively). Despite this, grain formation and maturity stages tended to support richer communities (up to 560 ASVs) compared to vegetative stages (480–500 ASVs). Shannon diversity followed a similar pattern, with higher values at maturity (4.6) and lower values during vegetative stages (4.1). Simpson indices indicated greater evenness at later growth stages (0.96) relative to earlier stages (0.91).

## Beta diversity of soil prokaryotic communities

Principal Coordinates Analysis (PCoA) based on Bray-Curtis dissimilarity revealed clustering of prokaryotic communities primarily by farming system and site, with secondary separation by crop growth stage (Fig 6). At the Chuka site season 1 and 2 the variation in prokaryotic diversity at sampling stages were not statistically significant, $p = 0.98$ and $p = 0.20$ whereas the variation in prokaryotic diversity in different treatments was statistically significant in season 1 and 2, $p = 0.01$ and $p = 0.01$, respectively. Principal Coordinates Analysis (PCoA) based on Bray-Curtis dissimilarities showed that at Chuka, the first two axes explained 47% of community variation (PCoA1: 34–37%; PCoA2: 13%). In the cereal season (Fig 6A) and potato season (Fig 6B), communities from later crop stages (grain formation and maturity) were separated from those at early vegetative stages. Farming system effects were also evident, with organic systems particularly Org-High forming distinct clusters away from conventional systems.

At Thika, a comparable pattern was observed, with the first two axes accounting for 40–46% of total variation. The variation in prokaryotic diversity at different sampling stages in seasons 1 and 2 were not statistically significant $p = 0.10$ and $p = 0.78$ whereas the variations in prokaryotic diversity in different treatments were statistically significant in season 1 and 2, $p = 0.001$ and $p = 0.001$, respectively. In both cereal (Fig 6C) and potato (Fig 6D) seasons, stage-based differences were apparent, as maturity-stage communities clustered apart from those in the vegetative stage. Farming system separation

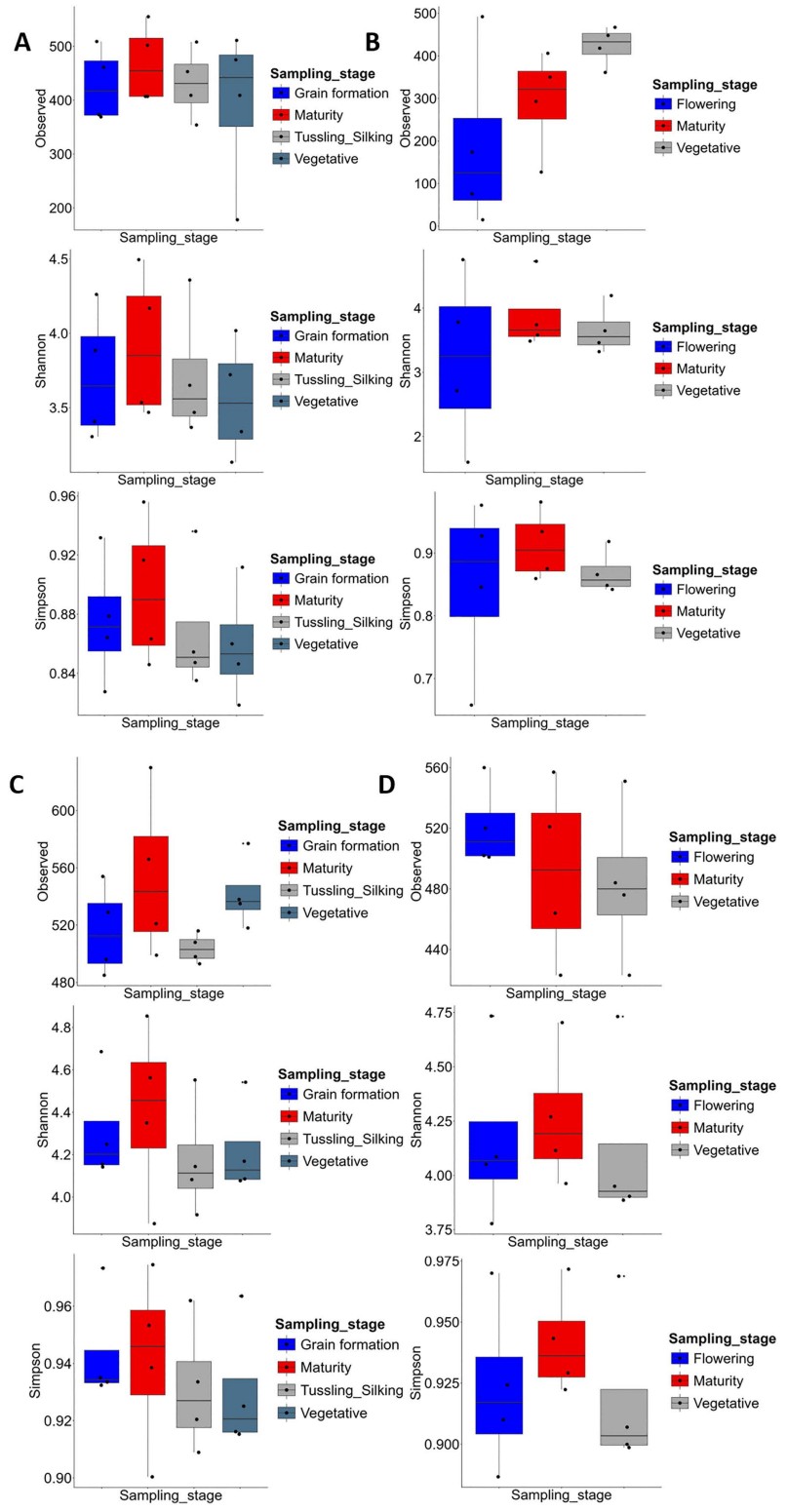

**Fig 5. Alpha diversity of soil prokaryotic communities across the different crop growth stages at Chuka and Thika sites.** Boxplots show observed amplicon sequence variants (ASVs), Shannon index, and Simpson index at different crop growth stages at Chuka (A, season 1; B, season 2) and Thika (C, season 1; D, season 2). Later crop growth stages (e.g., grain formation and maturity) generally exhibited higher richness and evenness compared to early stages.

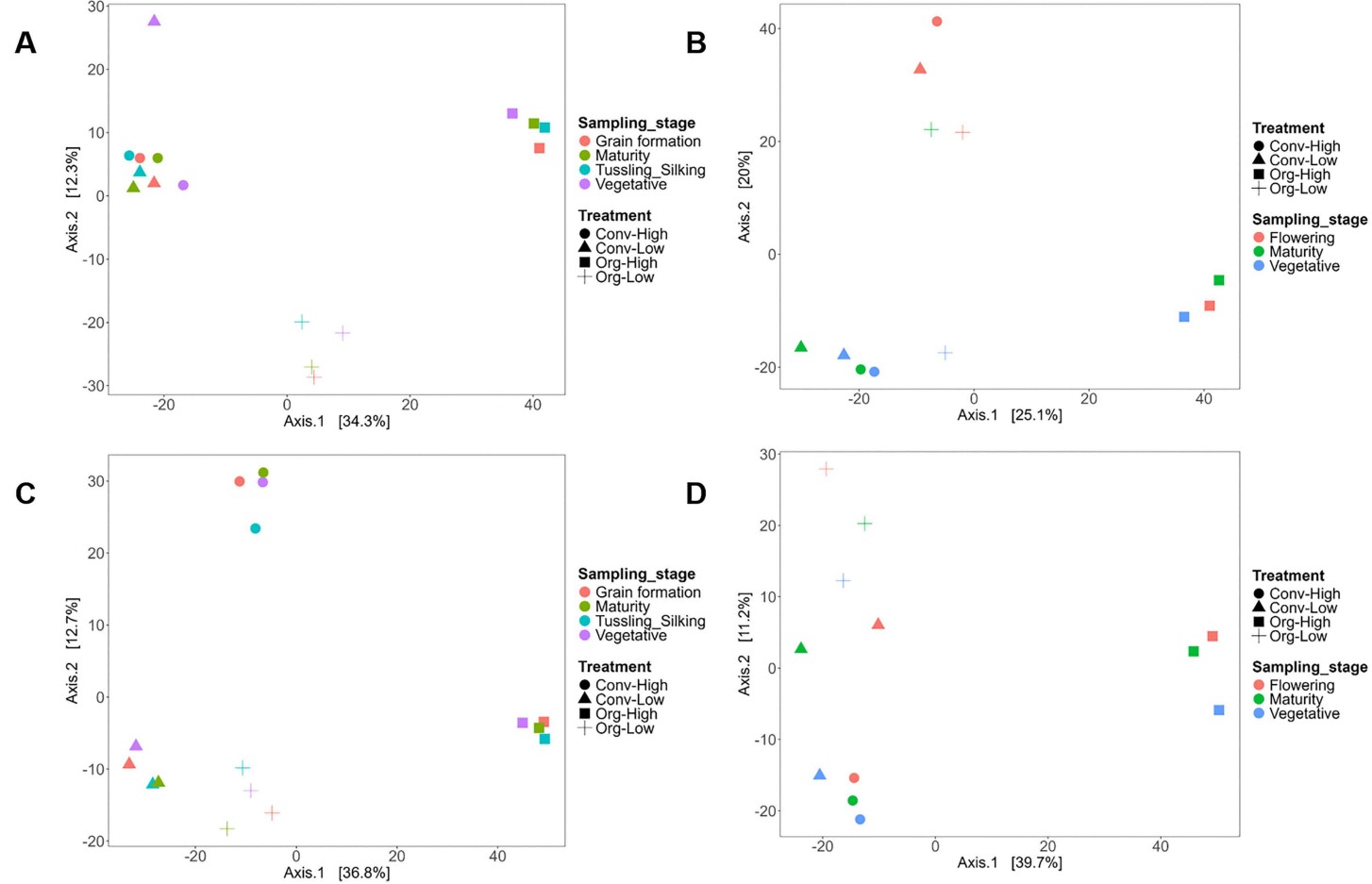

**Fig 6. Beta diversity of soil prokaryotic communities across farming systems at two sites.** Principal Coordinates Analysis (PCoA) based on Bray-Curtis dissimilarities illustrate community composition at Chuka (A, season 1; B, season 2) and Thika (C, season 1; D, season 2). Clustering patterns reflect differences among farming systems and crop growth stages.

was again clearest in the organic systems, especially Org-High, which showed greater divergence compared to conventional systems.

## Differential taxa across crop stages (LEfSe analysis)

The LEfSe analysis revealed differential enrichment patterns of selected prokaryotic genera across crop growth stages within farming systems. The heatmaps were based on log10 transformed relative abundances and highlight taxa contributing to compositional differentiation among vegetative, flowering/tasseling, grain formation, and maturity stages. (Fig 7). At Chuka (Fig 7A and 7B), enrichment of genera such as *Methanomassiliicoccus* and *Pseudoduganella* was associated with grain formation and vegetative stages, respectively, in the babycorn/maize season. In the potato season, additional taxa showed stage-specific enrichment, including *Zavarzinella* and *Limnoglobus* (vegetative stage), and *Pseudoflavitalea* and *Microvirga* (maturity).

At Thika (Fig 7C and 7D), *Lentzea*, *Sphingobacterium*, and *Novosphingobium* were more abundant during grain formation in babycorn/maize, while *Undibacterium* was linked to the vegetative stage. In the potato season, maturity stages

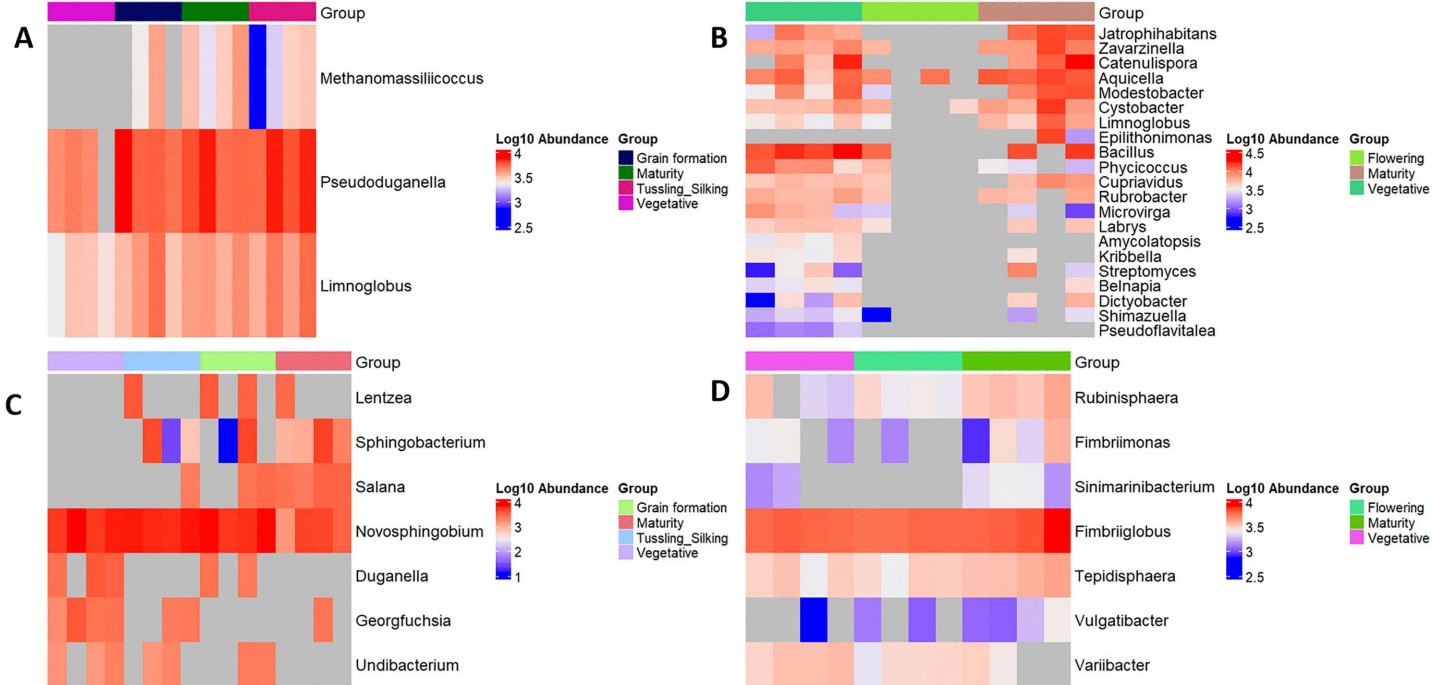

**Fig 7. Differentially abundant prokaryotic genera across crop growth stages and seasons.** Heatmaps based on Linear Discriminant Analysis Effect Size (LEfSe) show genera enriched at specific crop growth stages and seasons. Panels (A) and (B) represent Chuka (seasons 1 and 2), while panels (C) and (D) represent Thika (seasons 1 and 2). The log10 abundance scale indicates LDA scores, with colors marking the groups in which taxa were relatively more abundant. Given that LEfSe analyses were performed on relative abundance data derived from pooled samples, these results are interpreted as descriptive indicators of taxonomic differentiation across growth stages rather than statistically tested treatment effects or functional activity.

were associated with taxa such as *Rubinisphaera*, *Fimbriimonas*, and *Tepidisphaera*, whereas *Vulgatibacter* and *Variibacter* were more common at earlier growth stages.

## Discussion

This study provides insight into how long-term management intensity, organic input diversity, crop phenology, and site-specific soil conditions interact to shape soil prokaryotic communities in tropical agroecosystems. By revisiting the SysCom Kenya long-term experiment after 15 years of continuous management, we capture cumulative system effects that are not apparent in shorter-term studies and provide rare long-term microbial evidence from sub-Saharan Africa.

An earlier study from the same long-term trial by [14] reported higher prokaryotic diversity in conventional systems compared to organic systems. In contrast, the present study observed consistently higher richness and evenness in organic systems, particularly in the organic high-input treatment. These contrasting patterns likely reflect differences in temporal scale and system maturity. While the [14] study captured earlier phases of the experiment, the current analysis reflects 15 years of cumulative management, during which organic systems have received repeated applications of diverse organic inputs that may progressively enhance microbial habitat heterogeneity.

Methodological differences may also have contributed to divergence between the two studies. The present study employed an amplicon sequence variant (ASV)-based approach, which provides higher taxonomic resolution and greater reproducibility than operational taxonomic unit (OTU)-based clustering by resolving exact sequence variants at

single-nucleotide resolution [26,30]. Differences in bioinformatic pipelines can influence estimates of microbial richness and community structure and should therefore be considered when comparing results across studies.

Long-term application of organic amendments has been widely shown to increase microbial diversity, enzymatic activity, and community stability by enhancing soil organic matter heterogeneity and microhabitat formation [43,44]. In addition, differences in crop rotations, sampling seasons, and environmental conditions between sampling periods may have further contributed to contrasting outcomes. Taken together, these findings underscore that microbial community responses to farming systems are dynamic, and that patterns observed at earlier stages of long-term experiments may evolve as systems mature.

Although nitrogen form was the initial focus, our results suggest that organic input diversity and total nutrient input intensity are more informative descriptors of system differences than nitrogen form alone. All four systems received farmyard manure, but organic systems were distinguished by a wider diversity of organic amendments, including compost, *Tithonia diversifolia* mulch, plant teas, and legume intercrops. High-input systems also received substantially greater cumulative nutrient inputs over the crop rotation [14].

Organic input diversity promotes microbial richness by increasing the diversity of carbon substrates and creating a broader range of ecological niches. Previous studies have shown that heterogeneous organic inputs can reduce dominance by a few fast-growing taxa and enhance microbial evenness and resilience [45,46]. Our findings therefore support reframing system contrasts in terms of input diversity and long-term nutrient loading, rather than binary distinctions between organic and inorganic nitrogen sources [47].

Across sites and systems, prokaryotic diversity tended to increase from early vegetative stages toward grain formation and maturity. This pattern is consistent with increased rhizodeposition, root turnover, and residue inputs later in the crop cycle, which enhance substrate availability and microbial niche differentiation [48,49]. Similar stage-dependent increases in microbial diversity have been reported in maize and rice systems under field conditions [49].

However, phenological effects were not uniform across all systems, and variability among stages was high. Importantly, farming system effects were more consistent than phenological effects, indicating that long-term management exerts a stronger influence on microbial community structure than short-term temporal variation. Consequently, while crop growth stage contributes to temporal variability, it should be interpreted as a secondary driver relative to sustained system management.

Taxon-specific enrichment patterns identified using LEfSe further illustrate how crop growth stage and site context contribute to shifts in the relative abundance of individual prokaryotic genera. These patterns were variable across taxa and cropping systems, with no single genus consistently enriched across all stages or sites. Given that LEfSe analyses were based on relative abundance data derived from pooled samples, these results are interpreted as descriptive indicators of compositional differentiation rather than evidence of functional dominance or statistically tested treatment effects.

Canonical correspondence analysis revealed clear site-specific differences in how soil chemical variables were associated with microbial community structure. At Chuka, associations between measured soil variables and community composition were weak, whereas at Thika, pH, ammonium-N, and available phosphorus showed stronger relationships with community patterns.

The weaker associations at Chuka may reflect buffering effects of higher rainfall and moisture availability, which can reduce environmental filtering and dampen the influence of individual soil chemical parameters [50,51]. In contrast, the drier conditions at Thika likely amplify the role of pH and nutrient availability as selective pressures shaping microbial assemblages [52].

Soil organic carbon was not measured in this study, which limits interpretation of carbon-microbe interactions. Given the significant role of organic carbon in structuring microbial communities [53,54], its inclusion in future work would improve explanatory power, particularly at sites where soil chemical drivers appear weak.

Alpha diversity indices such as Shannon and Inverse Simpson provide useful summaries of richness and evenness but have important limitations. These metrics do not capture phylogenetic relatedness, functional redundancy, or microbial activity, and they are sensitive to sequencing depth, data processing choices, and compositional constraints inherent in amplicon sequencing data [29,34].

Additionally, microbial sequencing was conducted on pooled soil samples, precluding statistical inference at the replicate level. As a result, observed diversity patterns should be interpreted as indicative system-level trends rather than statistically tested treatment effects. While this approach limits inferential power, it enabled identification of broad patterns across a unique long-term experiment.

While taxonomic patterns provide insights into community structure, 16S rRNA gene sequencing does not directly resolve functional potential or microbial activity. Functional interpretations in this study are therefore inferential and based on known ecological roles of dominant taxa reported in the literature [40,41]. Future studies integrating replicate-level sequencing, shotgun metagenomics, metatranscriptomics, or enzyme activity measurements would be required to directly link microbial community composition with nutrient cycling processes.

Despite these limitations, the consistently higher richness and evenness observed in organic systems suggest greater microbial redundancy and potential resilience under variable environmental conditions, a key feature for sustainable soil management in smallholder systems [55].

Our findings indicate that long-term application of diverse organic inputs can foster richer and more even prokaryotic communities than conventional systems, particularly under tropical smallholder conditions. Aligning organic input management with site-specific constraints and recognizing the dominant influence of long-term system management over short-term temporal variation may enhance soil health and sustainability in sub-Saharan Africa.

## Conclusion

This study demonstrates that long-term farming system management, particularly the diversity and intensity of organic inputs, plays a significant role in shaping soil prokaryotic communities in tropical agroecosystems. After 15 years of continuous management in the SysCom Kenya long-term trial, organic systems especially those receiving higher and more diverse organic inputs consistently supported richer and more even prokaryotic communities than conventional systems across two contrasting sites in the Central Highlands of Kenya.

Across both sites, microbial diversity tended to increase toward later crop growth stages, reflecting temporal changes associated with rhizodeposition, root turnover, and residue inputs. However, these phenological patterns were variable and less consistent than differences attributable to long-term system management, indicating that sustained farming practices exert a stronger influence on microbial community structure than short-term crop developmental effects.

Soil pH, ammonium-N, and available phosphorus emerged as important abiotic correlates of community composition, particularly at the drier Thika site, highlighting the role of site-specific environmental filtering. Weaker associations at the more humid Chuka site suggest that climatic context can modulate the influence of soil chemical drivers. The absence of soil organic carbon measurements represents a limitation and should be addressed in future work.

Given the pooled sequencing design and the limitations of alpha diversity metrics, observed patterns should be interpreted as indicative system-level trends rather than statistically tested treatment effects. Nevertheless, this study provides rare long-term evidence from sub-Saharan Africa that diversified organic input management can enhance soil microbial diversity and potential resilience. Integrating composts, green manures, mulches, and legume-based intercrops into tropical farming systems may therefore contribute to improved soil biological health and long-term sustainability.

## Supporting information

**S1 Fig. Monthly climate variables at Chuka and Thika experimental sites.** Monthly rainfall (mm, bars), mean temperature (°C, brown solid line), and relative humidity (% RH, green solid line) recorded during the study period at (A) Chuka and (B) Thika sites.
(TIF)

**S2 Fig. Rarefaction curves of soil prokaryotic communities at Chuka and Thika sites.** Panels (A) and (B) represent the Chuka and Thika sites, respectively. Curves illustrate the relationship between sequencing depth and observed species richness, indicating that the sequencing effort provided sufficient coverage for robust downstream diversity analyses. (TIF)

**S1 Table. Tables A-F. Soil chemical characteristics across crop growth stages, farming systems, and sites in the SysCom Kenya long-term trial.** Table presents soil chemical properties measured at the Chuka and Thika sites under contrasting farming systems and cropping arrangements after long-term management. Tables **A** and **B** show the effects of crop developmental stages on soil pH, total nitrogen, nitrate-N, ammonium-N, Olsen phosphorus, and total phosphorus at Chuka and Thika, respectively. Tables **C** and **D** summarize soil chemical characteristics across organic and conventional farming systems within different cropping systems at Chuka and Thika. Tables **E** and **F** present the interaction effects between crop growth stage and farming system on soil chemical properties at Chuka and Thika, respectively. Values are means of replicate soil samples. Different letters within a column indicate significant differences among treatments where applicable. Significance levels are indicated as $p \leq 0.05$ (*), $p \leq 0.01$ (**), $p \leq 0.001$ (***), and ns denotes not significant. (DOCX)

## Acknowledgments

The authors thank the Kenya Agricultural and Livestock Research Organization (KALRO) for providing the trial site at Thika and Kiereni Primary School for hosting the trial site at Chuka. We are also grateful to Ms. Jane Makena and Mr. James Karanja for their dedicated management of the trial sites and assistance with data collection. We further acknowledge the contributions of the field and laboratory staff of the SysCom Kenya long-term trial for their support in soil sampling, data collection, and laboratory analyses. Finally, we thank the International Centre of Insect Physiology and Ecology (*icipe*) and the Research Institute of Organic Agriculture (FiBL) for their institutional support.

## Author contributions

**Conceptualization:** Susan Wairimu Muriuki, Anne Kambura, Edward Nderitu Karanja.

**Data curation:** Susan Wairimu Muriuki, Anne Kambura, Julius Mugweru, Kennedy Wanjau, Edwin Ngetha, Edward Nderitu Karanja.

**Formal analysis:** Susan Wairimu Muriuki, Anne Kambura, Kennedy Wanjau, Edwin Mwangi, Edwin Ngetha, Edward Nderitu Karanja.

**Funding acquisition:** David Bautze.

**Investigation:** Susan Wairimu Muriuki, Edward Nderitu Karanja.

**Methodology:** Susan Wairimu Muriuki, Anne Kambura, Julius Mugweru, David Bautze, Felix Matheri, Edwin Mwangi, Nancy Mwende, Edward Nderitu Karanja.

**Project administration:** Edward Nderitu Karanja.

**Supervision:** Anne Kambura, Julius Mugweru, Thomas Dubois, Edward Nderitu Karanja.

**Validation:** Kennedy Wanjau, Edward Nderitu Karanja.

**Visualization:** Susan Wairimu Muriuki, Anne Kambura, Kennedy Wanjau, Edward Nderitu Karanja.

**Writing – original draft:** Susan Wairimu Muriuki.

**Writing – review & editing:** Anne Kambura, Julius Mugweru, Kennedy Wanjau, David Bautze, Thomas Dubois, Felix Matheri, Edwin Mwangi, Nancy Mwende, Edwin Ngetha, Edward Nderitu Karanja.

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
