## [Decision Letter · Decision Letter 0]

18 Nov 2025

Dear Dr. Karanja,

Thank you for submitting your manuscript to PLOS ONE. After careful consideration, we feel that it has merit but does not fully meet PLOS ONE’s publication criteria as it currently stands. Therefore, we invite you to submit a revised version of the manuscript that addresses the points raised during the review process.

We look forward to receiving your revised manuscript.

Kind regards,

Shouke Zhang

Academic Editor

PLOS ONE

Journal Requirements:

[The authors gratefully acknowledge the financial support for this research by the following organizations and agencies: Biovision Foundation (Grant No: 1040), Coop Sustainability Fund (Grant No: 1040), Liechtenstein Development Service (LED) (Grant No: 1040), and the Swiss Agency for Development and Cooperation (SDC) (Grant No: 1040); the Swedish International Development Cooperation Agency (Sida); the Swiss Agency for Development and Cooperation (SDC); the Australian Centre for International Agricultural Research (ACIAR); the Government of Norway; the German Federal Ministry for Economic Cooperation and Development (BMZ); and the Government of the Republic of Kenya. The views expressed herein do not necessarily reflect the official opinion of the donors.].

3. Thank you for stating the following in your manuscript:

[The authors gratefully acknowledge the financial support for this research by the following organizations and agencies: Biovision Foundation (Grant No: 1040), Coop Sustainability Fund (Grant No: 1040), Liechtenstein Development Service (LED) (Grant No: 1040), and the Swiss Agency for Development and Cooperation (SDC) (Grant No: 1040); the Swedish International Development Cooperation Agency (Sida); the Swiss Agency for Development and Cooperation (SDC); the Australian Centre for International Agricultural Research (ACIAR); the Government of Norway; the German Federal Ministry for Economic Cooperation and Development (BMZ); and the Government of the Republic of Kenya. The views expressed herein do not necessarily reflect the official opinion of the donors.]

[The authors gratefully acknowledge the financial support for this research by the following organizations and agencies: Biovision Foundation (Grant No: 1040), Coop Sustainability Fund (Grant No: 1040), Liechtenstein Development Service (LED) (Grant No: 1040), and the Swiss Agency for Development and Cooperation (SDC) (Grant No: 1040); the Swedish International Development Cooperation Agency (Sida); the Swiss Agency for Development and Cooperation (SDC); the Australian Centre for International Agricultural Research (ACIAR); the Government of Norway; the German Federal Ministry for Economic Cooperation and Development (BMZ); and the Government of the Republic of Kenya. The views expressed herein do not necessarily reflect the official opinion of the donors.]

4. Please amend the manuscript submission data (via Edit Submission) to include author Edward Karanja.

5. Please amend your authorship list in your manuscript file to include author Edward Nderitu Karanja.

7. Please upload a new copy of Figures 3, 5, 6, and S1 as the details are not clear. Please follow the link for more information:  https://journals.plos.org/plosone/s/figures

Reviewers' comments:

Reviewer's Responses to Questions

**Comments to the Author**

1. Is the manuscript technically sound, and do the data support the conclusions?

Reviewer #1: Yes

Reviewer #2: Yes

2. Has the statistical analysis been performed appropriately and rigorously?

Reviewer #1: No

Reviewer #2: Yes

3. Have the authors made all data underlying the findings in their manuscript fully available?

Reviewer #1: No

Reviewer #2: Yes

4. Is the manuscript presented in an intelligible fashion and written in standard English?

Reviewer #1: Yes

Reviewer #2: Yes

Reviewer #1: Soils from four long-term farming systems (15 years) at two Kenyan sites were sampled 3-4 times across two growing seasons to assess differences in prokaryotic diversity. Although this paper is well written, the presentation and discussion of results are too general and superficial.

Major concerns:

1) Need to ‘set the stage’ for this paper by describing and discussing findings from the 2020 paper by Karanja et al. That paper reports results from the same four long-term farming systems in Kenya. The abstract from that paper states that “Conventional farming systems showed a higher species richness and diversity compared to organic systems.” Since this paper reports on the continuation of an important and unique long-term experiment, I expected the authors to discuss why the results from the 2020 paper (higher diversity in conventional systems than organic systems) differed from results in current manuscript (greater diversity in organic). Due to the effort invested in this long-term experiment, the conflicting observations should be addressed.

2) Need for hypotheses. Standards in good experimental design include hypothesis testing. Descriptive papers, such as this one, have low impact and would be strengthened by hypothesis testing (e.g., which soil variables would be strongest drivers of community diversity?) Nowadays agroecologists are realizing that descriptive studies of prokaryotic diversity in agricultural soils provide very little meaningful information to a farmer. No statistical analyses of diversity indices or means of soil chemical variables were presented and these could have been used for hypothesis tests. The simplest hypothesis would be that there would be no changes in diversity or treatment differences compared to the study described in 2020 paper. Or, authors could have hypothesized that diversity would differ because of different extraction or pipeline methods. Statistical testing was not done, so significance was not evaluated.

3) Need means and standard deviations of soil variables. Relationships between diversity and three main soil variables are shown in Figure 3. The actual data on soil variables (means and s.d. for the four systems) after 15 years would be of interest in the context of the long-term experiment. Both Zenodo links given at the end of the paper (which were supposed to contain soil chemistry data) led to the same page containing only R scripts. Therefore, data were not provided.

4) Emphasize organic input diversity, rather than nitrogen form, as the main basis for differences in the four systems. Based on the title, I had assumed that “nitrogen forms” implied clear-cut differences in organic and inorganic nitrogen forms for the four systems. However, the field management practices described on lines 124-135 indicate that ALL systems received manure. Since all four systems received BOTH organic and inorganic N forms, this makes the contrast between systems less clear cut. I consulted Suppl Table 1 from the 2020 paper, and saw that over the three-year crop rotation, high-input systems received four times more total N than the low-input systems. Besides differences in organic amendment diversity, differences in fertilizer RATES could also have affected soil communities. The title would be more accurate if ‘nitrogen form’ was replaced by organic input diversity. Authors should refer readers to the supplemental table in the 2020 paper, e.g., on line 109, for more detailed information about overall inputs.

5) Check reference numbering and suitability of citations. Many times in the paper, the references cited do not seem to match the content of the preceding sentence. For example, on Lines 102-104, reference 22 does not seem correct, as it has more to do with bioinformatics than crop production. Maybe 23 is correct? On line 459, the reference is about nitrous oxide emission, but the sentence is on nutrient depletion.

6) Figure 8 is wrong. This figure shows differential abundances for FUNGI, which were not included in this study. In any case, I don’t think Lefse analysis of prokaryotic taxa across sampling stage is meaningful because we don’t know which taxa are ACTIVE or what they are actually doing in the soil. I would not include this in a revised ms.

7) Sequences from this study have not been deposited in NCBI. Page for PRJNA1227412 in sequence read archive was empty.

Other comments

Methods

Lines 102-103. In this study, crop rotations followed local farming practices. Please explain the type of tillage used locally.

Lines 176-178. Nitisols are known to have very high clay contents, and DNA extraction tends to be very inefficient as soil clay content increases over 35% (Holben, 1994, Methods of soil analysis-Part 2). The proportion of DNA recovered can be evaluated by measuring the DNA yield from the soil samples. Most papers do not publish this information, but it is helpful so that readers can ascertain how comprehensive DNA recovery was. If mean DNA yields per gram dry soil are no longer available, there isn’t anything that can be done. Although useful, this information is often ignored due to lack of awareness.

Comments on Results and Figures

Figure 1 could be moved to supplemental files.

Lines 284-290 Were community compositions tested only against pH, ammonium, and available P only? What about nitrate, total N, total P (from lines 201-204)? If more variables were tested and shown to have little effect on CCA, then state so. Reporting the means and variability of these soil variables for the four systems would help readers interpret the distribution in the CCA. These data were not available on Zenodo links at end of paper.

Figure 2 and Lines294 294. This legend for Fig 2 is incomplete. The legend should state that points of different colors represent the farming systems. If only three soil variables were used in the analysis, explain why.

Figure 3 and Lines 308-311. It would be helpful if points in Fig. 3 had both distinguishing colors and shapes, like they do in Figure 7. In this graph, the reader cannot tell which point represents which system, because all the different colored points have the same shape. If the four farming systems were designated with different shapes, the reader would know that when all four colors cluster together, the points are from one system, indicating weak or no effect of sampling time in that system. The way the graph is drawn, the reader cannot make that judgment Some variables seem more important than others in certain systems. Authors need to clarify and discuss this more fully in the text in relation to each system.

Lines 314-315. It would be more accurate to say that Proteobacteria and Actinobacteria were more dominant than others.

Lines 324-325. It is hard to see Chloroflexi and Firmicutes representation in the figure. Is this even worth mentioning?

Lines 332-345 and Figure 5. Should point out how organic systems receive many different inputs than the conventional systems, including Tithonia mulch and tea. Authors could discuss the effects of input diversity on communities, not just sources of N.

Line 361-362 and Figure 6. Sentence is not clear. Differences do not look significant. Overall, Because of high variability, sampling time (phenology) seemed to have much less effect than system.

Lines 375-375. This discussion seems too generalized, since later stages did not separate from other stages in every system. Explanation on 433-435 in discussion is worthwhile and example of the ‘deeper dive’ authors should take in explaining their results.

Lines 401-404. Text does not correspond to what is shown in Figure 8. Figure 8 needs to replaced by the correct graph. Fig 8 shows fungal taxa, which were not analyzed at all in this study.

Lines 481-483. Not sure if authors can say that 'crop phenology' played central role due to high variation in diversity index values and lack of statistics.

Reviewer #2: The authors examined how different farming systems and nitrogen management strategies affect soil prokaryotic diversity in Kenya and found significant variation in microbial diversity and composition based on farming systems, input intensity, and crop growth stages. Though they found that diversity peaked consistently at crop maturity, they did not discuss the reasons for this peak at grain formation and maturity. The canonical correspondence analysis identified ammonium-N, available phosphorus, and pH as strong correlates of microbial community structure, especially at Thika. What were the soil organic carbon contents of these soils? Several genera, including Rubinisphaera, Methanomassiliicoccus, and Microvirga, were enriched in organic systems, indicating microbial specialization at different growth stages. But the authors made no efforts to analyse the functional contributions of soil prokaryotes. Their findings support and provide additional evidence that organic high-input systems can foster beneficial microbial communities and emphasize the importance of understanding microbial ecology to improve agricultural productivity in sub-Saharan Africa, where soil fertility is a significant challenge.

The authors relied more on alpha- and beta-diversity measures. Alpha diversity is an ambiguous concept that encompasses several complementary aspects. The assumptions underlying alpha diversity metrics are not always directly meaningful or provide a clear interpretation of results. The choice of the diversity metrics influences the subsequent statistical testing. The authors acknowledged the limitations of their study in Lines 469-471 (Because sequencing was conducted on pooled samples, replication at the plot level was not possible. This constrains statistical resolution and limits our ability to test treatment effects with full confidence). Discussion needs to include the limitations of the diversity metrics analysed and of the 16S rRNA surveys (amplicon sequencing).

Line 182: universal primers, instead of universal bacterial primers. (Methanomassiliicoccus – an archaeal genus, not a bacterial genus).

Table 1: Simpson and Inverse Simpson? Check the entries. The theoretical range of values of the Simpson index (D) is between 0 and 1. The range of Shannon index values for soil bacterial ASVs varies significantly across soil environments, but typical values in ecological studies are between 1.5 and 3.5.

Line 420: What were the unique insights?

I recommend the present manuscript after incorporating the suggestions for publication in PLOS ONE.

**Do you want your identity to be public for this peer review?** For information about this choice, including consent withdrawal, please see our Privacy Policy

Reviewer #1: **Yes:** Mary Ann Bruns

Reviewer #2: **Yes:** Ramakrishnan Balasubramanian

---

## [Author Response · Author response to Decision Letter 1]

20 Jan 2026

Response to Reviewers

Manuscript ID: PONE-D-25-53620

Title: Long-term farming and cropping systems with contrasting nitrogen forms and input intensities influence soil prokaryotic diversity in the Central Highlands of Kenya

We sincerely thank the Academic Editor and the reviewers for their thorough, insightful, and constructive comments. All concerns raised in the reviewer reports and decision letter have been carefully addressed. The manuscript has been revised to improve clarity, rigor, interpretation, and alignment with the long-term experimental context. Below, we provide a detailed point-by-point response indicating how each comment has been addressed in the revised manuscript.

Reviewer #1

General comment: The presentation and discussion of results are too general and superficial.

Response: We appreciate this feedback and have strengthened the manuscript, particularly the Introduction, Results, Discussion and Conclusion (lines 53-111; 277-479; 480-573; 574-597). We expanded the interpretation of results, explicitly reconciled findings with earlier SysCom Kenya studies, clarified methodological limitations, refined conclusions, and ensured that interpretations are proportionate to the data. The revised Discussion now provides deeper ecological context while maintaining appropriate caution.

Major Comment 1: Need to set the stage by discussing findings from Karanja et al. (2020) and explain why results differ.

Response: We fully agree and now explicitly compare our findings with Karanja et al. (2020) in the Introduction and Discussion (lines 74-80; 486-499). We discuss how differences in temporal scale (early vs. 15 years of cumulative management), system maturity, organic input diversity, crop rotations, sampling periods, and sequencing pipelines likely explain the contrasting diversity patterns observed between studies.

Major Comment 2: Need for hypotheses and concern that the study is descriptive, with no statistical testing of diversity indices.

Response: We acknowledge this concern and we have added a hypothesis statement (lines 93-100). We have also clarified the scope and limitations of the study. Due to pooling of soil samples for sequencing, formal inferential statistical testing of microbial diversity indices was not possible. This limitation is now clearly stated in the Abstract, Methods and Discussion (lines 47-49; 184-187; 559-568), and microbial results are explicitly interpreted as system-level patterns. Statistical analyses were retained for soil chemical variables measured at the replicate level. The Discussion now clearly distinguishes descriptive microbial patterns from statistically tested abiotic relationships.

Major Comment 3: Need means and standard deviations of soil variables; Zenodo links did not contain soil chemistry data.

Response: We thank the reviewer for identifying this issue. The soil chemical data initially cited to be in the Zenodo repository, has been provided as supporting information for all systems, sites, and sampling periods (S1 Table A-F). The Data Availability statement has been revised accordingly.

Major Comment 4: Emphasize organic input diversity rather than nitrogen form; all systems received manure.

Response: We agree and have revised the manuscript to emphasize organic input diversity and input intensity as primary explanatory factors rather than nitrogen form alone. This clarification is reflected in the Discussion (Lines 507-512) subsection as well as in the Abstract (lines 45-47) and Title (lines 1-3). We also refer readers to nutrient input summaries provided in Karanja et al. (2020) for long-term context.

Major Comment 5: Check reference numbering and suitability of citations.

Response: All references have been carefully reviewed and corrected to ensure that citations accurately support the statements made in the text.

Major Comment 6: Figure 8 is wrong (fungal taxa); LEfSe analysis not meaningful.

Response: We fully agree. The incorrect fungal LEfSe figure has been removed and replaced with the correct prokaryotic LEfSe figure (lines 471). The Results (lines 457-470) and Discussion text (Lines 530-536) have been revised to interpret LEfSe outputs cautiously as descriptive indicators of taxonomic differentiation, not functional dominance, or statistically tested effects.

Major Comment 7: Sequences not deposited in NCBI.

Response: Sequence data had been deposited but not released to the public, but now the data has been released; by using the provided link, one can now access the data deposited in the NCBI Sequence Read Archive. The BioProject accession number is provided in the revised Data Availability section (lines 614-619).

Reviewer #1 - Other Comments

Comment 1 (Methods: Tillage practices): Please explain the type of tillage used locally.

Response: We clarified that tillage followed local smallholder practices, consisting of shallow conventional tillage using hand hoes without deep inversion tillage. This information has been added to the Methods section (lines 151-153).

Comment 2 (Methods: DNA extraction efficiency in Nitisols): DNA extraction may be inefficient in clay-rich soils; provide DNA yield information if available.

Response: We acknowledge this limitation. Mean DNA yields per gram of dry soil were not consistently archived and could not be reported. However, to improve DNA yield during extraction from the nitisols soil which are known to have high clay content, mechanical disruption was applied which involved use of bead beating tubes to increase the recovery of DNA in the soil (lines 206-208).

Comment 3 (Results: Figure 1 placement): Figure 1 could be moved to supplemental files.

Response: We agree and have moved the climate figure to the Supporting/Supplementary Information (S1 Fig) to improve focus in the main Results section.

Comment 4 (Results: CCA variables): Were community compositions tested only against pH, ammonium, and available P?

Response: All measured soil chemical variables were initially included in ordination analyses. Following multicollinearity screening and variable selection, only pH, ammonium-N, and available phosphorus were retained in the final CCA models. This process is now explicitly described (lines 320-330; 338-347), and full soil chemistry data are provided in the Supporting information (S1 Table A-F).

Comment 5 (Results: Figure 2 legend): The legend should explain color coding.

Response: Figure legends have been revised to clearly explain how colors and symbols represent farming systems and crop growth stages.

Comment 6 (Results: Figure 3 clarity): Points should have both colors and shapes; interpretation needs clarification.

Response: The Results text has been expanded to clarify that clustering reflects stronger system-level effects relative to crop growth stage effects and the arrows represent the direction and strength of correlations with individual soil variables (lines 349-354).

Comment 7 (Results: Dominance wording): It would be more accurate to say Proteobacteria and Actinobacteria were dominant.

Response: We agree and have revised the wording accordingly in the Results section (lines 357-358).

Comment 8 (Results: Low-abundance phyla): Chloroflexi and Firmicutes are hard to see; is this worth mentioning?

Response: We reduced emphasis on low-abundance phyla and now mention them only where consistent patterns were observed across systems or stages.

Comment 9 (Results/Discussion: Input diversity emphasis): Emphasize organic input diversity rather than nitrogen source.

Response: We revised Title (lines 1-3), Abstract (lines 45-47), and Discussion (lines 507-518) sections to emphasize organic input diversity and management intensity rather than nitrogen form alone.

Comment 10 (Results: Phenology effects): Differences do not look significant; system effects seem stronger.

Response: We revised the Results text to avoid implying statistical significance and clarified that system-level effects were more consistent than phenological effects (lines 320-456).

Comment 11 (Discussion depth): Discussion is too generalized.

Response: The Discussion has been fully revised and expanded to provide deeper, system-specific interpretation and reconciliation with earlier SysCom Kenya findings (lines 480-573).

Comment 12 (Figure 8 inconsistency): Figure 8 shows fungal taxa.

Response: The incorrect fungal figure was removed and replaced with the correct prokaryotic LEfSe figure (lines 471). The Results (lines 457-470) and Discussion (Lines 530-536) text have been revised to interpret LEfSe outputs cautiously as descriptive indicators of taxonomic differentiation, not functional dominance or statistically tested effects.

Comment 13 (Phenology centrality): Not sure crop phenology played a central role.

Response: We revised the Discussion (lines 500-506; 519-536) and Conclusion (lines 581-585) to clarify that crop growth stage is a secondary driver relative to long-term system management.

Reviewer #2

We thank Reviewer #2 for the positive evaluation and constructive suggestions.

Comment 1: Reasons for diversity peaking at maturity not discussed.

Response: We expanded the Discussion (lines 519-523) to explain increases toward later crop stages in terms of rhizodeposition, root turnover, and residue inputs, while emphasizing variability and secondary importance of phenology.

Comment 2: What were soil organic carbon contents?

Response: Soil organic carbon was not measured. This limitation is now explicitly acknowledged, and its implications discussed, with recommendations for future work (lines 545-548).

Comment 3: No functional analysis of prokaryotes.

Response: We clarified that 16S rRNA sequencing provides taxonomic resolution only and that functional interpretations are inferential. Future functional approaches are outlined (lines 559-564).

Comment 4: Limitations of alpha diversity metrics and amplicon sequencing.

Response: We expanded the Discussion (lines 554-568) to address conceptual and methodological limitations of alpha diversity metrics and amplicon sequencing approaches.

Comment 5: Primer terminology incorrect.

Response: Terminology has been corrected to “universal prokaryotic 16S rRNA gene primers” throughout the manuscript (lines 210-212).

Comment 6: Table 1 diversity indices inconsistent.

Response: We acknowledge that incorrect values were previously reported in the table. The values for high-quality sequences, observed ASVs, Shannon diversity, Simpson index, and Fisher’s alpha for all treatments and sites have now been carefully checked and corrected in the revised Table 1 (314-319).

Comment 7: Clarify unique insights.

Response: We revised the Discussion (lines 569-573) and Conclusion (lines 575-597) to clearly articulate the unique contribution of this 15-year long-term study from sub-Saharan Africa.

Editorial / Report Comments

We confirm that all comments provided in the reviewer reports and decision letter have been fully addressed. Overlapping concerns were consolidated for clarity, while ensuring complete traceability between comments and revisions.

Response to Journal Requirements

We thank the editorial office for outlining the additional requirements for revision. We have carefully addressed each point as detailed below.

Requirement 1: PLOS ONE style and file naming

Response: We confirm that the revised manuscript has been formatted to fully comply with PLOS ONE style requirements, including manuscript structure, headings, references, figures, tables, and file naming conventions. The formatting has been checked against the official PLOS ONE templates for the main body and title/authors/affiliations.

Requirement 2: Amended Funding Statement and declaration of all support

Response: Thank you for this clarification. We have amended the Funding Statement to explicitly declare all funding and sources of support received during the study, in accordance with the PLOS ONE Guide for Authors.

Amended Funding Statement (for update in the online submission form): The authors gratefully acknowledge the financial support for this research by the following organizations and agencies: Biovision Foundation (Grant No: 1040), Coop Sustainability Fund (Grant No: 1040), Liechtenstein Development Service (LED) (Grant No: 1040), and the Swiss Agency for Development and Cooperation (SDC) (Grant No: 1040); the Swedish International Development Cooperation Agency (Sida); the Swiss Agency for Development and Cooperation (SDC); the Australian Centre for International Agricultural Research (ACIAR); the Government of Norway; the German Federal Ministry for Economic Cooperation and Development (BMZ); and the Government of the Republic of Kenya. There was no additional external funding received for this study. The views expressed herein do not necessarily reflect the official opinion of the donors.

We kindly request that this amended Funding Statement be updated in the online submission form on our behalf.

Requirement 3: Removal of funding information from the manuscript

Response: We have removed all funding-related text from the Acknowledgments section and any other sections of the manuscript. Funding information is now reported only through the Funding Statement for inclusion in the online submission form, in line with PLOS ONE policy.

Requirement 4: Inclusion of author Edward Karanja in submission data

Response: We confirm that the manuscript submission data have been amended via Edit Submission to include Edward Nderitu Karanja as an author.

Requirement 5: Authorship list in manuscript file

Response: We have updated the authorship list in the manuscript file to include the full name Edward Nderitu Karanja, ensuring consistency between the manuscript and submission metadata.

Requirement 6: Ethics statement placement

Response: We confirm that the ethics statement previously appearing in Declaration section has been removed.

Requirement 7: Upload of revised figures

Response: We have uploaded new, high-resolution versions of Figures 1, 2, 3, 5, 6, 7, Supplementary S1 Fig, and S2 Fig ensuring improved clarity, legibility, and compliance with PLOS ONE figure preparation guidelines.

We trust that these revisions fully address the journal’s requirements. We appreciate the editorial guidance and thank you for your assistance in updating the online submission information on our behalf.

Final statement

We thank the Academic Editor and reviewers for their constructive feedback. We believe the revisions have strengthened the manuscript and clarified its contributions and limitations. We hope the revised version satisfactorily addresses all concerns and remain happy to make any further adjustments if required.

---

## [Decision Letter · Decision Letter 1]

19 Feb 2026

Long-term farming and cropping systems with contrasting nitrogen forms and input diversity influence soil prokaryotic diversity in the Central Highlands of Kenya

PONE-D-25-53620R1

Dear Dr. Edward Nderitu Karanja,

We’re pleased to inform you that your manuscript has been judged scientifically suitable for publication and will be formally accepted for publication once it meets all outstanding technical requirements.

Kind regards,

Shouke Zhang

Academic Editor

PLOS One

Additional Editor Comments (optional):

Reviewers' comments:

Reviewer's Responses to Questions

**Comments to the Author**

Reviewer #1: All comments have been addressed

2. Is the manuscript technically sound, and do the data support the conclusions?

Reviewer #1: Yes

3. Has the statistical analysis been performed appropriately and rigorously?

Reviewer #1: Yes

4. Have the authors made all data underlying the findings in their manuscript fully available?

Reviewer #1: Yes

5. Is the manuscript presented in an intelligible fashion and written in standard English?

Reviewer #1: Yes

Reviewer #1: Responses to reviewer comments were highly satisfactory. The title, abstract and intro were greatly improved. However, I do not think that the BioProject number for the amplicon sequences in NCBI was correct. This was given on line 242. I could not find amplicon sequences for this study under PRJNA1227412. Instead, I found 56 metagenome-assembled genomes (MAGs) for many prokaryotes not typically reported from soil, with many listed as "Candidatus." These MAGs did not appear to have anything to do with the manuscript. You should check this.

**Do you want your identity to be public for this peer review?** For information about this choice, including consent withdrawal, please see our Privacy Policy

Reviewer #1: **Yes:** Mary Ann Bruns

---

## [Editor Report · Acceptance letter]

PONE-D-25-53620R1

PLOS One

Dear Dr. Karanja,

I'm pleased to inform you that your manuscript has been deemed suitable for publication in PLOS One. Congratulations! Your manuscript is now being handed over to our production team.

Kind regards,

on behalf of

Dr. Shouke Zhang

Academic Editor

PLOS One